# Strain versus Tunable Terahertz Nanogap Width: A Simple Formula and a Trench below

**DOI:** 10.3390/nano13182526

**Published:** 2023-09-09

**Authors:** Hwanhee Kim, Mahsa Haddadi Moghaddam, Zhihao Wang, Sunghwan Kim, Dukhyung Lee, Hyosim Yang, Myongsoo Jee, Daehwan Park, Dai-Sik Kim

**Affiliations:** 1Department of Physics, Ulsan National Institute of Science and Technology (UNIST), Ulsan 44919, Republic of Korea; psikvv@unist.ac.kr (H.K.);; 2Quantum Republic Co., Ltd., Rm 805-6 Bldg 106, UNIST-gil, Eonyang-eup, Ulju-gun, Ulsan 44919, Republic of Korea

**Keywords:** nanogap, tunable zerogap, terahertz, diffraction pattern, nano-trench, gaptronics, pet substrate

## Abstract

A flexible zerogap metallic structure is periodically formed, healing metal cracks on a flexible substrate. Zerogap is continuously tunable from nearly zero to one hundred nanometers by applying compressive strains on the flexible substrate. However, there have been few studies on how the gap width is related to the strain and periodicity, nor the mechanism of tunability itself. Here, based on atomic force microscopy (AFM) measurements, we found that 200 nm-deep nano-trenches are periodically generated on the polymer substrate below the zerogap owing to the strain singularities extant between the first and the second metallic deposition layers. Terahertz and visible transmission properties are consistent with this picture whereby the outer-bending polyethylene terephthalate (PET) substrate controls the gap size linearly with the inverse of the radius of the curvature.

## 1. Introduction

Terahertz science and technology is making rapid advances in imaging [1,2,3,4,5,6,7,8,9], sensing [10,11,12,13,14,15,16,17], and quantum optics [18,19,20]. Recent studies involve using such plasmonic structures as tip and gap, resulting in significant field enhancements for nano-spectroscopy [21,22,23,24,25,26,27,28,29]. Increasingly, creating a narrow gap and controlling that gap are of paramount importance.

Nanogap plasmonics are being investigated across a broad spectrum, spanning from the visible light to the terahertz regime. Near-field enhancement in nanogaps of various types leads to surface-enhanced Raman scattering (SERS) and strong light–matter interaction [30,31,32,33,34,35,36,37,38,39]. This nanogap platform can also be used for constructing long wavelength polarizers, both negative and positive, which are related to the vector Babinet’s principle [40,41,42,43]. Decreasing the gap size makes higher field enhancement until the gap size becomes much smaller than the metallic film thickness; when the periodicity becomes much smaller than the wavelength; and when small gaps introduce quantum contacts between two metal planes [42,44,45,46,47,48,49,50,51,52,53].

Light-matter interaction, especially of the field enhancement, can be described by the capacitor model, which proves to be useful for many applications such as sensor, filter and switch [41,54,55,56,57,58,59,60], where quantum plasmonics probed by terahertz time domain spectroscopy (THz-TDS) for large aspect ratio nanogap is of prime interest [61]. In contrast to the fixed or limited tunability of nanogaps used in most studies, our group has pioneered the zerogap technology [41,42,62]. This advancement enables active control of gap width, allowing adjustment from nearly zero to several hundred nanometers. We construct the zerogap structure on a flexible polyethylene terephthalate (PET) substrate. While this method is not trivial and it takes a few days to manufacture one wafer, in essence, it enables hundreds of samples of different gap widths to be integrated into one sample, making it a unique and versatile platform.

Despite these recent developments [41,62], an intuitive approach connecting the gap width to other relevant experimental parameters, such as the periodicity and strain or radius of curvature, is lacking. In addition, exactly what happens to the polymers in the vicinity of the gap is also unknown. In this paper, we investigate the relationship between the gap width and strain for our Au zerogap samples of 5 μm periodicity. Using AFM, gap sizes are measured as we outer-bend the sample. To unravel how the PET structure below the Au zerogap is modified, we etch out the top Au layer and probe morphology of the exposed surface of PET. For the bended substrate, we find that deep trenches appear periodically, suggesting that periodic Au nanogaps are involved in the formation of them. After repeated bending cycles, we believe that gold films in the vicinity of the zerogap become essentially free-standing, detached from the polymer substrate. This results in a surprisingly simple formula connecting strain and gap width, as well as tunable terahertz field enhancement, which has a well-defined maximum as a function of the gap width.

## 2. Materials and Methods

### 2.1. Materials Sample Fabrication

First, a gold film with a thickness of 100 nm [63] is evaporated onto a 250 μm thick commercial PET (polyethylene terephthalate) using an electron beam evaporation system, which is used as a flexible substrate [41,62]. On top of the gold film, microscale arrays are patterned with standard photolithography using a photoresist layer by spin coating with AZ5214E (AZ Electronic Materials, Luxembourg), then soft baked at 95 °C and irradiated with UV light 62 mJ cm^−2^ in a mask aligner (MDA-400S, MIDAS SYSTEM, Daejeon, Republic of Korea), located under a photomask with the microscale chromium patterns of slits with different periodicities (1:1 line/space, a periodicity of 5 μm). After baking the exposed photoresist at 115 °C, the sample is irradiated for a second time with UV light 210 mJ cm^−2^ without photomask. The microscale pattern becomes visible after 50 s of development of the photoresist pattern in AZ300MIF (AZ Electronic Materials) and acts as a shadow mask for the ion milling process. After etching the unprotected part of the gold layer with the photoresist pattern by an argon ion beam (Ar) at an angle of incidence of 0°, a second gold layer of the same thickness is deposited on the entire structure. In this case, where an optimal bond between gold and the substrate must be achieved, a 3 nm thick titanium adhesive layer is evaporated immediately before the deposition of the two gold layers. Finally, the gold trenches with zero-gap junctions are obtained by immersing the sample in NMP solvent at 90° for 2 h to remove the photoresist layer.

### 2.2. Methods

#### 2.2.1. AFM Information and Measurements

The main system for the AFM scanning is Park NX20 (Park Systems, Suwon, Republic of Korea) and the tip is OMCL-AC160TS (Park Systems, Suwon, Republic of Korea). The tip radius is 7 nm, and the tip side angle is 18 degrees. AFM measurement mode is the tapping mode with a set point of 15 nm and an amplitude of 29 nm.

In these experiments, the sample bending holder is made by the x-axis micrometer stage. The sample bending holder is attached to the silicon wafer to fit the specifications of the Park NX20. The detailed experimental procedures are as follows: We remove the Park NX20 header. The AC160TS tip is attached to the header using the magnetic holder of the tip and the header is fastened to the main set-up. We adjust the reflected laser beam off the cantilever, so that it is positioned at the center of the CCD. The bent sample is xy-positioned to the AFM tip. Frequency sweeping of the cantilever tip enables the resonance frequency determination, and the set-point is chosen to be 15 nm away from the sample surface. After a rough scan of 20 μm (x = 256 px) × 20 μm (y = 64 px) within 5 min to find the approximate sample morphology, we zoom onto a 500 nm × 500 nm area in the vicinity of the opened nanogap created by bending.

#### 2.2.2. Time Domain Spectroscopy (TDS) Terahertz Measurements

Mode-locked femtosecond Ti:Sapphire 80 MHz repetition rate laser pulses split into two, a pump beam and a probe beam. The pump beam illuminates a DC-biased gallium arsenide THz emitter. This gallium arsenide THz emitter generates a pulse with a Fourier-transformed spectrum center at 0.5 THz. The off-axis parabolic mirror focused the THz pulse on the sample. The spot diameter of the THz pulse is around 3.5 mm. The sample is held on a bending stage. The effective arc length of the sample is 17.4 mm. The transmitted THz pulse from the sample was focused to 1 mm thick (110) orientation zinc telluride by an off-axis parabolic mirror. The probe beam goes through the delay stage to control the path difference between the THz pump and the 800 nm wavelength probe beam. The varying delay time of the probe beam enables time domain spectroscopy measurement. The probe experiences slight polarization rotation owing to the presence of the terahertz beam inside the 1 mm-thick (110) zinc telluride crystal, which in turn is used to detect the terahertz electric field amplitude using λ/4 wave plate, Wollaston prism, and balanced detector.

The THz pules pass through a 250 μm PET substrate and then the zerogap at the transverse magnetic (TM) mode incidence. All THz-TDS settings are covered by nitrogen gas purging to reduce the humidity of less than 10% because the THz transmission rate is affected by the humidity [64,65]. We change the curvature of the sample by the bending stage to measure the THz transmission amplitude of the bent zerogap sample. The reference THz transmission amplitude is defined as that through the bare 250 μm thickness PET sample. We rotate the bending stage to measure the TE mode with the same procedure as the TM mode. Fast Fourier Transform (FFT) converts the measured THz pulse to a frequency domain.

#### 2.2.3. 532 nm Laser Beam Transmission

To measure the diffraction pattern of the sample, a 532.1 nm Ti:Sapphire laser Millennia Pro (Spectra Physics, Milpitas, CA, USA) was used. By employing convex and concave lenses with a laser beam, we achieved a beam diameter of 1.4 mm at the 1/e^2^ points. The beam goes to the PET substrate and the zerogap. The power meter quantified the diffraction pattern orders emanating from the zerogap sample. We adjust the sample stage from the strain 0% to 2.6% to vary the sample gap width. We measured three cycles of bending and releasing and took the average. The bare PET transmission intensity is 40 mW.

#### 2.2.4. Gold Etchant Procedure

After being cleaned with acetone sonication for 10 min, the sample is immersed in sigma aldrich gold etchant (standard) for one hour and then in H_2_O_2_ for another hour, after which it is rinsed in water.

#### 2.2.5. Simulation

We used COMSOL Multiphysics 6.1 with a wave optics module. We applied a Drude model (ω_p_ = 13.7 PHz, γ = 40.7 THz) to gold and the refractive index of PET was assumed to be 1.78. To obtain the conductance-gap size relation, the dielectric constant given by Equation (2) was assumed in the gap region for each gap size and we searched the value of an effective conductivity of the gap region, which makes the simulated transmission equal to the experimental value.

## 3. Results

### 3.1. Surface Morphology of Flexible Gap

We successfully fabricate 5 μm period zerogap device on a PET substrate. To confirm that our zerogap sample has tunable gap width, we bend and release the device to check light transmission through a variable nanogap. For this experiment, we load the zerogap onto the strain stage, as depicted in Figure 1a. In this configuration, the stage holds each edge of the device, and we bend it by moving one edge toward another fixed one. To check whether our sample works well as designed, we monitor the center of zerogap while bending the sample. The effective arc length of the sample is fixed at 17.4 μm. The reflection and transmission optical microscope images of a flat sample are shown, respectively, in Figure 1b,c, where the nominal 5 mm periodicity is shown. It should be noted that light transmission through the gap on a flat surface is only slightly stronger than the direct transmission, because the typical gap widths are only 2–3 nm when the sample is flat, with many hidden Ohmic connections [42,62]. Figure 1d shows a bent sample with the absolute displacement value of the sample dx. Usually, the dx range is typically from 0 to 1.5 mm. The reflection and transmission optical microscope images of a bent sample with dx = 1.5 mm and a radius of curvature in the middle of 5 mm are shown, respectively, in Figure 1e,f. It should be noted that because of the curvature, the outer rims of the sample are already out of focus. Nevertheless, far stronger transmission through the gap indicates that under the outer-bending gaps opened up significantly.

### 3.2. AFM Measurements, Nano-Trench

When the micrometer stage bent the sample with a displacement of dx = 1 mm, the strain of the PET in the middle is around 1.8% at which point a gap width of 76 nm is seen (Figure 2a). The strain is estimated with Equation (1). With twice of the radius of the curvature of the AFM tip added (7 nm × 2 = 14 nm), it results in a 90 nm estimated gap width. Incidentally, this gap width is almost exactly the same as the strain multiplied by the periodicity, which leads us to make a conjecture about the relationship between the gap width (ω) and the strain:(1)$$Gap\;width\;≅\;curvature\;\cdot \;\frac{h_{sub}}{2}\;\cdot \;period=\varepsilon P_{0},$$
curvature = 1/r_c_ where r_c_ is the radius of curvature of the PET substrate at the center, and the strain ε is given by *h_sub_*/(2r_c_). This trend has already been predicted by one of our earlier works [66]. To see if this is indeed the case, at least phenomenologically, we change the strain and perform the gap width measurement using AFM. Indeed, the measured gap width precisely follows the simplistic phenomenological Equation (1) (Figure 2b). Strain singularities are formed where the gaps are located [66,67,68], which would exert enormous force on the PET below the Au gap at which nonlinear tearing will occur, generating trenches effectively canceling the initial singularity [69,70]. This trench will smooth out the opening and closing of the nanogap structure, ensuring repeatability of the performances [41], forming free-standing gaps on top of the PET trench to ensure the phenomenological validity of Equation (1). To verify this hypothesis, we etched away the gold film and removed the top zerogap structure, allowing us to examine the PET substrate itself. Figure 2c displays an AFM scan image measuring within the area of 20 μm by 20 μm with dx = 1 mm, aimed at investigating our hypothesis that a nano-trench exists beneath the zero-gap. This trench is expected to facilitate both the smooth opening and closing of the gap. Figure 2d is the remeasure of the specific point to find detailed information on the nano-trench. Figure 2e shows the line profile of the topography in Figure 2d, to check the depth of the trench, set at 0 for the PET surface. It shows a depth of around 200 nm, twice the thickness of the gold film before etching. The topography of x displacement of 900 mm (Strain 1.79%) is shown in Appendix A.

### 3.3. Terahertz Transmission Experiments

In metallic nanogap structures, the transmission coefficient of the terahertz radiation depends on the gap size of the sample. Figure 3a shows a schematic of the THz transmission measurements. The Transverse Magnetic (TM) mode has its polarization perpendicular to the zerogap lines. The Transverse Electric (TE) mode has polarization parallel to the zerogap line. The expected gap size is proportional to the curvature. In Equation (1), the sample period *P*_0_ is 5 μm, $h_{sub}$ is the thickness of the substrate equal to 250 μm, and *ε* is the strain. Figure 3b shows the radius of curvature at the center, with the hypothetical neutral plane dissecting the middle point of the plane with zero strain. For the TM mode transmission THz TDS measurements, Figure 3c shows the different transmission amplitudes depending on the curvature of the substrate. The largest curvature of the sample (the smallest radius of curvature) reaches almost 100% transmission over the entire spectral range, consistent with the results of a higher density gap sample of smaller width (200 nm periods, 5 nm gap width) [71].

As the sample becomes flat, the gap width is reduced to zero and the device becomes optically equivalent to gold film with the same thickness, as the transmission approaches the direct transmission through bare Au film. To verify our experimental results, we simulate the Terahertz transmission rate for each polarization. Figure 3d (blue line) shows the result of the 0.5 THz TM mode transmission simulation, where the transmission quickly reaches unity as the gap width increases. The black and green solid lines represent the experimental results of the 0.5 THz transmission coefficient for the TM and TE modes, respectively. The discrepancy between these data could be associated with the existence of numerous local contacts made by the salient features of the uneven respective metal layers, generating an imaginary part of the dielectric constant [42,46,66,72].
(2)$$\varepsilon_{r}\left(\omega \right)=1+\frac{\sigma_{total}}{\varepsilon_{0}\omega }\;i$$

The imaginary part of the permittivity attenuates terahertz waves and it was shown that the imaginary part of unity is the critical point that determines whether significant field enhancement and large transmission is possible [31]. Thereby, the estimated conductance per the RC model of ref [42] is shown in Figure 3e. According to the gap size, period and transmission rate, we can find Field Enhancement (*FE*).
(3)$$FE=\frac{E}{E_{0}}\cdot \frac{P_{0}+w}{w}$$
*E*/*E*_0_ is simply the normalized transmission amplitude, *P*_0_ is the sample period 5 μm, and w is the gap width. The *FE* graph is shown in Figure 3f. Owing to the effective conductance of the gap, the graph exhibits a peak when the gap width is 62 nm. We note that, had there been no quantum contacts between the adjacent metal planes, the field enhancement would show a monotonic decrease with increased gap with strain.

### 3.4. Zerogap 532 nm Laser Beam Diffraction Pattern

As the microwave transmission coefficient of zerogap is well explained in previous research, we checked a visible wavelength 532 nm laser beam transmission (Figure 4a) [41]. Figure 4b shows the diffraction pattern camera images according to the sample bending displacements (dx), demonstrating good sample quality over macroscopic length scale; indeed, these diffraction patterns constitute the best means by which we can readily verify the sample quality. The TM and TE modes 0th, 1st and 2nd order diffraction pattern intensities are shown in Figure 4c–h. The measurements have some hysteresis depending on whether in bending or in the release cycle. The black line is the bending procedure, and the red line corresponds to the release procedure. There is an intensity difference at the same sample stage displacement for bending and release. We assume that this happens because of the sample stage back-lash.

## 4. Discussion

In this paper, we investigated various aspects of tunable metallic nanogaps, with an emphasis on how the gap width changes with strain. This would give a back-of-the-envelope way of estimating the gap quickly without having to perform time-consuming AFM, SEM or TEM scans each time. The surprisingly simple relationship between the gap width and the strain, mediated by the periodicity, is as intuitive as the railroad gaps designed to accommodate thermal expansion and shrinkage of rails during summer and winter months. The deeper-than-metal film depth trenches imply that indeed, the gap width can be much larger than the 100 nanometer range of this work, even up to 1000 nanometers or beyond, if we can apply more strains with softer materials such as Polydimethylsiloxane (PDMS) and using larger periodicity. The application potential of our zerogap technology already include biological systems because our gap widths span molecular, protein and virus sizes. With further strain, we might be able to encompass human cells with our wide-open gaps, again closing down to zero. This ability for nano and microfluidics applications is important for nanohole DNA sequencing, for instance. Since our nanogap structure is essentially TEM waveguide mode-like, with no cut-offs, visible, far-infrared or longer wavelengths can be easily described by a simple circuit model combined with quantum conductance arising from conducting channels, as our RC model and deduction of the effective conductance has demonstrated. It should be noted that measuring the effective conductance of the partially conducting nanogap was only possible through the application of terahertz time domain spectroscopy. In particular, extended gaps are expected to greatly enhance the research activities of long wavelength photonics, including terahertz spectroscopy. Continuously adjusting the size of the gaps from the zero gap to hundreds of nanometers in repeatable fashion will have far-reaching implications for the science and engineering of optics and photonics.

## 5. Conclusions

We verified the electromagnetic wave transmission characteristics of 5 μm period flexible zerogap plasmonic structure across a wide range, from the terahertz to the visible light region. Based on the AFM measurements, we contend that the formation of nano-trenches beneath the zerogap is an important aspect of our zerogap structure. From the strain and period of the gold zerogap structure, we discovered a simple gap width formula. Through terahertz pulse transmission, we were able to deduce the field enhancement. By means of Drude model simulations, we could ascertain the effective complex dielectric constant of the actual gap. This increasing conductivity of the gap with decreasing width explains why the sample attains a maximum field enhancement value at a specific strain. Furthermore, we were able to verify the terahertz polarization dependence through the TM and TE modes of this zerogap structure transmission experiments. This zerogap platform isn’t confined solely to the terahertz range. Rather, it enables the verification of various transmission characteristics across microwaves, as well as visible light. This zerogap is anticipated to find applications not only in sensing, near-field, and wave modulation, but also in detecting subtle strains, enabling the monitoring of bodily movements or precise mechanical motions.

## Figures and Tables

**Figure 1 nanomaterials-13-02526-f001:**
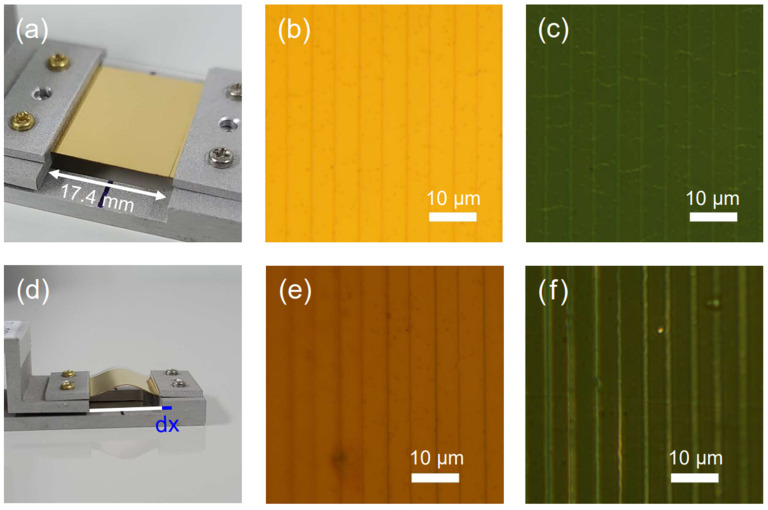
Zerogap sample and stage image information. (**a**,**d**) shows the zerogap sample image with the micrometer stage. The effective arc length of the sample is 17.4 mm. dx is the absolute value of the sample displacements shown in the blue line in (**d**). (**b**,**c**) is the normal 5 μm period 100× optical microscope image of reflection and transmission illumination. The white line on the image is the scale bar, which is 10 μm (**e**,**f**) is the 2.04 cm^−1^ radius of curvature (dx = 1.5 mm) of the 100× optical microscope image of reflection and transmission illumination. The outer rims of the samples are defocused because of the sample radius of curvature. The white line on the image is the scale bar, which is 10 μm. (**c**,**f**) have the same illumination and exposure time of 2 s.

**Figure 2 nanomaterials-13-02526-f002:**
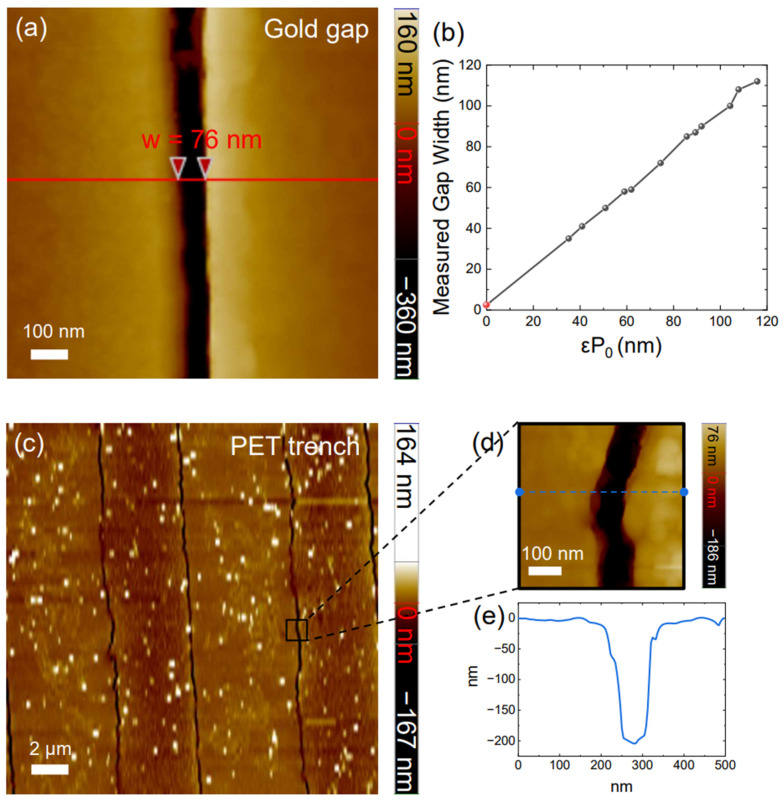
AFM topography of the bent zerogap sample and PET trench after gold removal. (**a**) shows the gold gap topography of the bent zerogap in 1.8% strain (radius of the curvature is 1.47 cm^−1^). The width of the gap is 76 nm. The white line is the 100 nm scale bar. The topography depth scale is on the right side of the topography. The strain multiplied period of the sample versus the measured gap size is shown in (**b**). *ε* is the strain, and *P*_0_ is the period. In this experiment, the period is 5 μm. The y-axis value of the red dot is 2.5 nm [62]. (**c**) shows the 20 μm × 20 μm size topography zerogap after etching the gold in 1.8% strain (radius of the curvature is 1.47 cm^−1^). The white spot is expected to be the remaining gold particles. There are periodic nano-trench lines. The white line is the 2 μm scale bar. The topography depth scale is on the right side of the topography. (**d**) depicts a close-up view of one of the trenches (**c**). The size of the topography is 500 nm × 500 nm. The nano-trench is located in the center. The white line is the scale bar and the depth scale is on the right side of the topography. (**e**) is the line profile along the blue dotted line in (**d**). The line sets the surface of the PET sample as 0 nm.

**Figure 3 nanomaterials-13-02526-f003:**
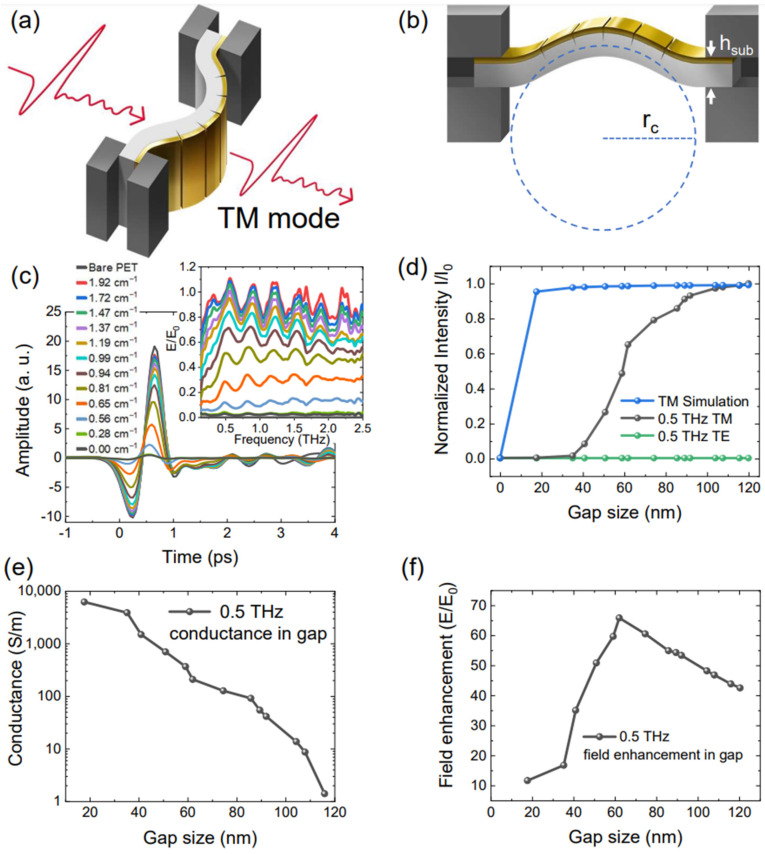
The Terahertz transmission experiments. (**a**) shows the schematic of Terahertz TM mode transmission experiments. (**b**) shows the radius of the curvature in the sample center. The dashed line represents the neutral plane, which is the middle point of the plane with zero strain. r_c_ is the sample radius of the curvature and *h_sub_* is the sample thickness. The TDS-THz transmission amplitude graph is (**c**). The results in the frequency domain are presented in the top right inset. The lines are the curvature from 0.00 cm^−1^ to 1.92 cm^−1^ and bare PET. (**d**) shows the comparison of the TM mode simulation, TM mode experiments, and TE mode experiment of 0.5 THz normalized transmission intensity. (**e**) The RC model conductance graph from the result (**d**). The terahertz field enhancements are shown in (**f**).

**Figure 4 nanomaterials-13-02526-f004:**
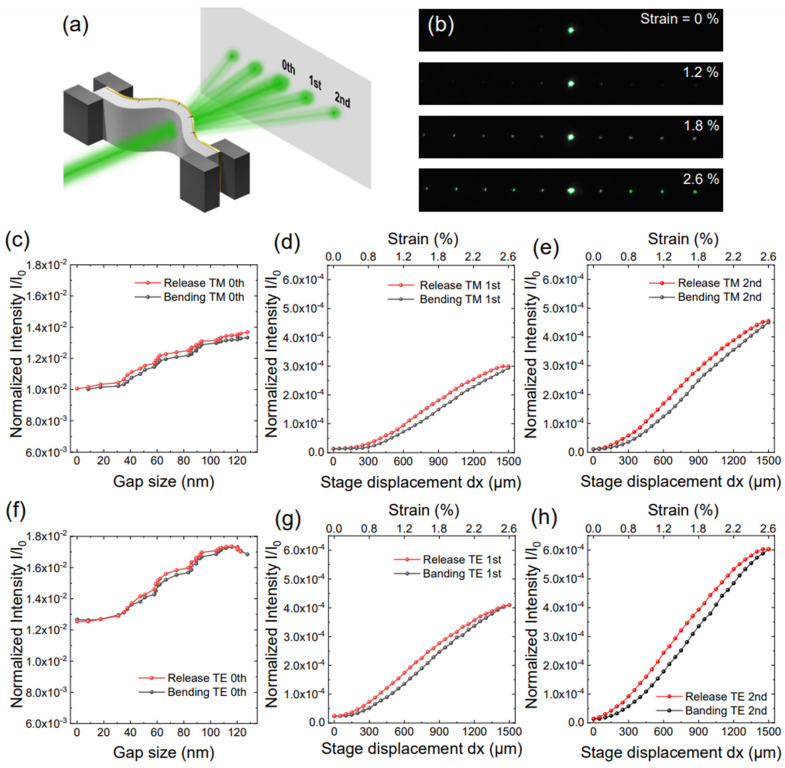
Visible laser beam diffraction pattern of 532 nm wavelength. (**a**) Schematic of the 532 nm laser beam diffraction pattern. (**b**) shows the 532 nm green laser diffraction pattern through the zerogap sample. The TM mode 0th order normalized transmission intensity is shown in (**c**). The TM mode 1st and 2nd order normalized transmission intensity is shown in (**d**,**e**). The TE mode 0th order normalized intensity is shown in (**f**). The TE mode 1st and 2nd order normalized intensity is shown in (**g**,**h**).

## Data Availability

Not applicable.

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
