# Peer review of "Strain versus Tunable Terahertz Nanogap Width: A Simple Formula and a Trench below"

_nanomaterials, 2023, doi:10.3390/nano13182526_

Round 1

Reviewer 1 Report

The manuscript presents about strain versus tunable terahertz nanogap width. Following are my comments.

1.       Mention some of the mathematical results in the abstract section like AFM measurement data. The comparison statement in abstract section is not clear. It needs revision.

2.       What are the difference values ranges for real experimental data and present some elaborative explanation in introduction section?

3.       Material and methods needs great improvement. Its not presented in well manner.  

4.       Explain in detail for each sub figures of Figure 1 in more descriptive form.

5.       Elaborate more in figure 4. It requires more description.

6.       Discussion section is not well arranged.

7.       Conclusion and comparison table is missing in manuscript.

Overall, paper needs heavy revision. Do revise the paper as suggested.

paper needs extensive revision, please. 

Author Response

We thank referee 1 for careful review of our paper. We revised the main contents based on the reviewers' comments:

  1. We have included the information on the 200 nm-deep nano-trenches observed in AFM measurements in the abstract section.
  2. We have incorporated additional contents related to terahertz time domain experiments in the Introduction.
  3. We have provided a detailed experimental methodology in the Methods section.
  4. We have provided more detailed information about Figure 1 in Result 3-1.
  5. We have incorporated a content regarding the 532 nm green laser transmission experiment for the flexible zerogap in Result 3-4.

     6, 7. We have added a conclusion section to summarize the overall content of the paper

In the present organization of the manuscript, we humbly contend that a comparison table is not essential.

We thank the referee again for reading this paper carefully; your comments have greatly enhanced the quality of this paper.

Reviewer 2 Report

The paper reports on a gold film deposited onto a flexible (PET) substrate. So-called zero-gap trenches are inscribed on the system: thanks to the application of a strain, obtained by controlled mechanical deformation of the system, trenches are opened and optical properties modified, in particular in the THz regime (occurrence of a diffraction pattern in the visible, also briefly reported in the paper, appears as a quite trivial result).

There is not much originality nor in results or their discussion, since the basic idea was already demonstrated by the same team (see, e.g., Ref. 8) and explored, in more or less modified versions, by other groups. Focus of the present paper is the topographical analysis of the trench width, carried out by AFM on both the whole system and the underlying substrate, after gold, and the resulting claimed agreement with a simple formula relating width with strain via the inscribed periodicity.

Despite its limited originality, the paper is of interest and can be definitely considered for publication. Prior to acceptance, I suggest Authors to consider the following points and improve the manuscript, accordingly.

1.     Style and language should be carefully revised all through the text in order to improve readability.

2.     I found both Introduction and Conclusions not very clear not particularly appropriate. In the Introduction, Authors confusedly mention (and put together) effects and phenomena that, while possibly sharing some “near-field”-based mechanism, do pertain to different fields and, mostly, different wavelength ranges. In the Conclusions, they open the possibility to use the system as a sensor, in the substance. I think that: (i) Introduction should be more tightly tied to the effective topic of the paper, that is, in my opinion, the realization of a tunable-transmission filter, with obvious polarizing effects, to be used in the THz, or far-IR, range; (ii) the idea put forward in the Conclusions, to use the system for sensing applications, would certainly require more discussions aimed, in particular, at pointing out any potential advantage of the proposed system. Also, since deformation investigated in the paper are not small, in absolute terms, the feasibility of a practical device employing, e.g., solutions, as customary made in biosensing, should be briefly addressed (would concentration remain unchanged upon deformation?)

3.     The experimental part lacks some details. Although I understand most of the required info can be retrieved in the past literature by the same team, I think the reader would find useful to know, for instance, how the system is patterned (this is not mentioned at all in the present paper), how the AFM investigation is made on the bent sample, why transmission in the THz range is investigated through TDS measurements.

4.     I am always very suspicious of any scanning probe microscopy investigation which is apparently carried over single sample features. As a matter of fact, measurements should be repeated in different portions of the sample (e.g., in different positions along the same trench and in different trenches) and some minimal statistical approach used to determine relevant data. In other words, I think that the plot in Fig. 2(b) (but also 3(d) and 3(e)) should appear with adequate error bars to reflect the unavoidable spread in gap thickness in the fabricated samples.  

Language and style must be carefully revised, including choice of terms, sometimes not adequate for a scientific publication.

Author Response

We thank referee 2; much of the main contents have been revised based on the reviewers' comments. Regarding novelty or lack thereof, in previous studies, the existence of trenches was not observed, nor considered. In addition, the explicit gap width formula as a function of the strain was not given. These two contents constitute the main leap of this paper relative to earlier works [41, 62]. In addition, the existence of maximum field enhancement point as a function of the gap width is revealed in this paper explicitly for the first time. We thank the referee for this critical improvement on our paper.

1. To enhance readability, various aspects of the contents including introduction, conclusion and methodology have been enhanced.

2. We have incorporated additional contents into the introduction section pertaining to the paper's experimental focus on terahertz aspects. Reflecting feedback from other reviewers, we have also introduced additional information regarding 532 nm laser transmission experiments applicable in the context of a quick sample quality check. We also add contents on biosensing in the discussion section, highlighting the potential of transferring these zerogap structures onto different substrates to achieve much larger gap sizes.

3. We have included additional information about the sample and the experimental setup in the Methods section. Regarding the bent sample, at a curvature of 0.147 mm^-1 and an AFM scan range of 20 μm, there is a maximum height difference of 29 nm. At a scan range of 500 nm, there is a difference of only 0.01 nm center-to-end, indicating a near-flat nature of the sample within the scan. However, due to the flexibility of the substrate, caution should be added in the tip approach.

4. We include topography data from the same experiment conducted at different curvatures in the supplementary data in the same sample but at different locations. We consider that the sample discrepancies are explained through the imaginary part of the dielectric constant. The relevant results can be found in Figure 3 (e).

Reviewer 3 Report

I can recommend this paper for publication. However, authors should explain in more details: 1. Simulations, THz experiments and present transmission spectra. 2. Please use mixing formulas for material parameters of system. It gives you explanation why your system transmit THz radiation in this manner. You can find similar approach as in following papers:

https://doi.org/10.3390/ma16051948

DOI 10.1088/1361-6528/acb712

DOI 10.1088/1361-6528/ac7403

Author Response

We thank referee 3 for the careful review. We have added details about the simulation to the Methods section. The explanation for the imaginary part of the dielectric constant was provided through the previous study [42,46,66]. Thank you for recommending these relevant papers; we have cited all three of them.

Reviewer 4 Report

Comsol findings and experimental results should be bechmarked. Errors in between should be analyzed and discussed. This section should be very detailed.

4.Discussion acts like the Conclusions of this study. 4.Discussion should report the findings of the experimental works.

A Conclusion section should be added to report the core results of this entire work study.

Author Response

We thank referee 4. The explanation for the imaginary part of the dielectric constant was provided through the previous study [42,46,66]. This explains the difference between simulation and THz measurements. We have added a conclusion section to present the outcomes of the entire research.

Round 2

Reviewer 1 Report

Just rearrange the section of conclusion. It should be in last part of manuscript. Other is good. 

Its fine

Author Response

We appreciate referee 1 for taking the time to review the revised paper once again. We have rearranged the order of the conclusion and discussion sections. We also welcome any additional comments at any time. Thank you.

Reviewer 4 Report

Overall, very good revisions.

1) Conclusion cannot have any reference!
2) Discussion should be placed before Conclusion.

Author Response

We appreciate referee 4 for taking the time to review the revised paper once again. We have removed all references from the conclusion section and switched the positions of the conclusion and discussion sections. Thank you for helping improve the overall quality of the paper.